# Moving towards Biofuels and High-Value Products through Phytoremediation and Biocatalytic Processes

Elena Ionata [1,†], Emilia Caputo [2,†], Luigi Mandrich [1,*] and Loredana Marcolongo [1]

1 Research Institute on Terrestrial Ecosystems-IRET-CNR, Via Pietro Castellino 111, 80131 Naples, Italy; elena.ionata@cnr.it (E.I.); loredana.marcolongo@cnr.it (L.M.)
2 Institute of Genetics and Biophysics-IGB-CNR, "A. Buzzati-Traverso", Via Pietro Castellino 111, 80131 Naples, Italy; emilia.caputo@igb.cnr.it
* Correspondence: luigi.mandrich@cnr.it
† These authors contributed equally to this work.

**Abstract:** Phytoremediation is an eco-friendly technology that utilizes plants and plant–microbe interactions to remove a wide spectrum of organic and inorganic pollutants from contaminated environments such as soils, waters and sediments. This low-impact, environmentally sustainable and cost-effective methodology represents a valuable alternative to expensive physical and chemical approaches, characterized by secondary pollution risks, and is gaining increasing attention from researchers and popular acceptance. In this review, the main mechanisms underlying the decontamination activity of plants have been clarified, highlighting the environmental remediation in fertility and soil health. Studies have illustrated the high potential of phytoremediation coupled with green and sustainable biocatalytic processes, which together represent a non-polluting alternative for the conversion of plant biomass into renewable resources. The convenience of this technology also lies in the valorization of the bio-wastes towards biofuels, energy purposes and value-added products, contributing to an effective and sustainable circular approach to phyto-management. The strategy proposed in this work allows, with the use of totally green technologies, the recovery and valorization of contaminated soil and, at the same time, the production of bioenergy with high efficiency, within the framework of international programs for the development of the circular economy and the reduction of greenhouse carbon emissions.

**Keywords:** phytoremediation; bio-waste; biocatalytic processes; biofuel; integrated techniques

## 1. Introduction

In recent years, rapid urbanization and industrialization and the enormous impact of anthropogenic activities have begun to represent a serious concern for the environment, with numerous toxic substances indiscriminately dumped and the absorption capacity of natural ecosystems and the dilution effect relied upon to remove them. Indeed, their accumulation is seriously affecting the entire biosphere, threatening the safety of biota and human health through biomagnification along food chains [1]. Polychlorinated biphenyls (PCBs), polycyclic aromatic hydrocarbons (PAHs), chlorinated solvents, pesticides, antibiotics, heavy metals (HMs) and radionuclides are among the organic and inorganic compounds that cause serious environmental pollution. They are released through untreated sewage, runoff and effluents from intensive agricultural and industrial activities, including mineral extraction, petroleum refinery, and chemical manufacturing. In Europe, according to European Environmental Agency, polluting activities are distributed over large areas in which approximately 250,000 heavily contaminated sites have thus far been identified, which in turn will prospectively become approximately 1,500,000 by 2025 [2]. Soils, waters (wastewaters, surface and underground waters) and sediments require urgent remediation intervention and the greatest challenge of our societies is the development of increasingly adequate decontamination strategies by which to save the planet's health.

Huge economic efforts have been afforded by industrialized countries; in America alone, 6–8 billion dollars per year have been spent on the cleanup of polluted areas, with total investments so far amounting to around 25–30 billion dollars [3]. In Europe it is estimated that, by 2048, funding of an estimated 100 trillion euros will be allocated by the European Community for bioremediation of polluted areas [3]. Various physical and chemical approaches have been attempted, but high costs have hindered their application [4]. Moreover, they entail numerous hazards and risks for the safety of ecosystems, such as the alteration of the structures of autochthonous microbial communities and the generation of secondary pollution [5]. Thus, bioremediation strategies, based on the exploitation of biological systems such as plants, animals [6] and microorganisms [7], which achieve the removal or transformation of toxic pollutants into less or completely non-hazardous forms, represent low-impact, sustainable and cost-effective techniques, and are considered the most feasible strategies for the restoration of polluted sites [8].

In recent years, phytoremediation, which exploits plants and their specific processes, metabolisms and plant–microbe interactions, has been increasingly implemented and applied, gaining much interest among scientists and popular acceptance [9]. Environmental remediation by green plants involves the absorption, translocation, accumulation, and transformation of contaminants through metabolic activities into components of plant tissues or their mineralization to $CO_2$. It is important to underline that the limitation of pollutants mobility (stabilization), which hinders their leaching into aquifers, is also achieved through phytoremediation interventions, leading to important improvements in soil health, fertility and usefulness [10]. Over the past decades, scientists have extensively studied the genetic and physiological mechanisms underlying plant response and resistance to toxic compounds. Evaluations of different phytoremediation strategies have been previously conducted in controlled environments on single pollutants, in species-specific experimental trials [11]. The ideal plants for decontamination purposes should be able to survive in harsh and highly contaminated natural environments and produce high biomass. Further, these plants should be tolerant to the toxic effects of contaminants, easy to grow, characterized by an efficient root absorption and be inedible for herbivores [12]. In many cases, aiding interventions are necessary to achieve effective environment decontamination. Different studies have shown that microbial-assisted phytostimulation by exploitating bacterial endophytes or mycorrhyzal associations is very effective when improving the efficiency of phytodegradation. Moreover, the use of biostimulants and amendments (composted sewage sludge, biochar and EDTA) to improve soil properties, or genetic engineering approaches to ameliorate plant remediation capacities, are commonly used aiding strategies [13].

The main objective of this overview, after having clarified the main mechanisms underlying the decontamination activity of plants, is to demonstrate the high potential of phytoremediation. The convenience of this strategy, being 5 to 13 times cheaper than other techniques, also lies in the exploitability of the biomass produced for biofuels and energy purposes, biofortified products, and carbon sequestrations, which contribute to a circular approach to phyto-management.

The novelty of this review is the description of "sustainable phytoremediation" strategies developed by coupling the restoration of contaminated marginal areas with the utilization of high-biomass crops for bio-energetic purposes. This allows the recovery and valorization of degraded lands toward the high-yield production of biofuels in the frame of circular economy programs and of the abatement of greenhouse carbon emissions. Several studies on the utilization of post remediation bio-wastes for biofuels and bioenergy production through biological green and environmentally friendly methodologies are discussed. An in-depth investigation of biomass pretreatment methods was carried out highlighting their importance not only in reducing the lignin barrier recalcitrance but also in promoting the release of HMs from the biomass. This is the basis of clean bio-fuel production and metal recovery aimed at the integration of phytoremediation with biomass biorefinery towards minimum waste generation. Then, we undertake an extensive analysis of several

bio-processes that, through the utilization of suitable biocatalysts (enzymes and microorganisms), allow post-phytoremediation residue exploitation for biofuels production. A particular focus was placed on properly designed strategies that couple phytoremediation interventions and biocatalytic methodologies for biomass conversion into liquid or gaseous biofuels. Different studies are described that are centered on the selection of the most suitable plant species, the most effective bio-processes configurations, conditions and the choice of the adequate biocatalysts for maximizing biofuel yields with zero waste outputs and no risk of secondary pollution.

## 2. Phytoremediation: Mechanisms and Plant Selection

There are several ways by which plants clean up or remediate contaminated sites (Figure 1). The uptake of contaminants in plants occurs through the root system, in which the main toxicity prevention mechanisms are activated. The roots system provides a large surface area that absorbs and stores water and nutrients essential for growth along with other non-essential contaminants.

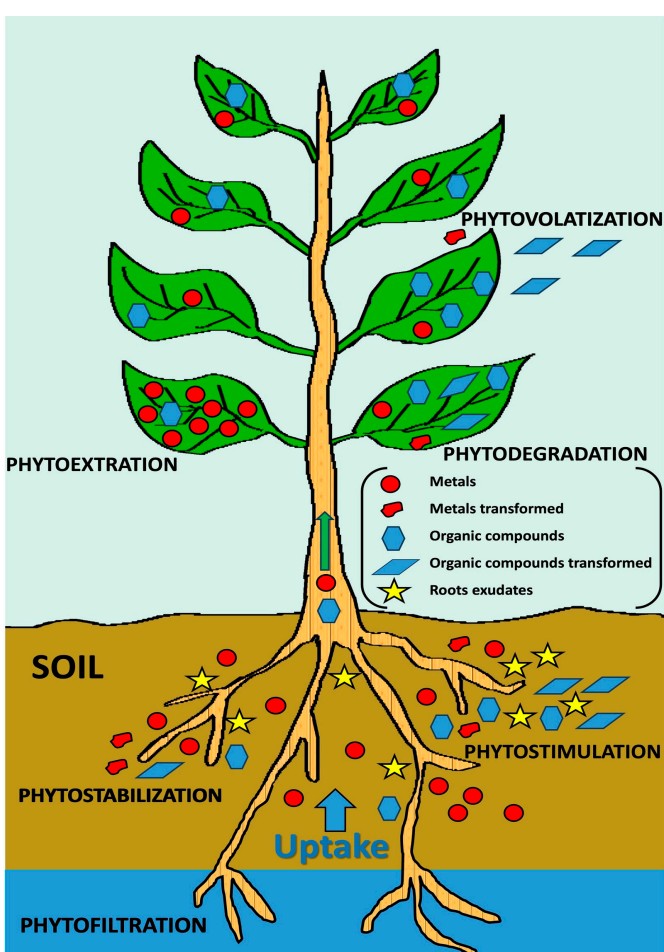

**Figure 1.** Schematic representation of phytoremediation. Phytofiltration, sequestration in root cells of contaminants from surface and/or ground water; phytostabilization, limitation of mobility and availability of pollutants in rhizosphere by roots; phytostimulation, degradation/transformation of organic compounds by plant root exudates and symbiotic microorganisms in the rhizosphere; phytodegradation, breakdown or transformation of contaminants by enzymes within vegetal tissue; phytoextraction, accumulation of metals in aboveground tissue (shoots, leaves, stalks); phytovolatilization, conversion of metals into volatile forms and release into atmosphere through leaf surface.

Phytoremediation methods and efficiency depend on the type of contaminant, bioavailability, and environmental properties. As reported in Table 1, different plants exploit

specific mechanisms or combinations to decontaminate the environment, e.g., plants can accumulate metals in their leaves and shoots and degrade organic toxic substances in their above-ground tissue or rhizosphere by root associated microorganisms [9].

**Table 1.** Plants for phytoremediation and the different mechanisms utilized by each species.

| Species | Life Cycle | Mechanism | Contaminants | References |
|---|---|---|---|---|
| *Populus* spp. | Perennial | Phystimulation, Phytodegradation Phytofiltration Phytostabilization Phytovolatilization | [a] TCE, Atrazine, [b] PCBs, [c] HCH Pb, Ni, V, Cr, Sn, Pb [d] TCA | [14–16] [17] [16] [14] |
| *Cannabis sativa* | Annual | Phytoextraction | Ni, Pd, Cd | [18] |
| *Brassica napus* | Annual | Phytoextraction Phytovolatilization | Cd, Cr, Cu, Ni, Pb, Zn Se | [19] [20] |
| *Brassica juncea* | Annual | Phytoextraction Phytovolatilization Phytofiltration | Cd, Pb Hg | [21] [22] |
| *Medicago sativa* | Perennial | Phytovolatilization Phytodegradation | TCA | [23] |
| *Ulmus glabra* | Perennial | Phytostabilization | As, Cu, Cr, Ni, Pb | [24] |
| *Pteris vittata* | Perennial | Phytoextraction | As | [25] |
| *Sedum alfredii* | Perennial | Phytoextraction | Zn, Cd, As, Pd, Cu | [26] |
| *Alyssum bertolonii* | Perennial | Phytoextraction | Ni, Co | [27] |
| *Thlaspi caerulescens* | Perennial | Phytoextraction | Pb, Cd, Ni, Zn, Co, Mn | [28] |
| *Sylibum marianus* | Annual | Phytoextraction | Cd | [29] |
| *Miscanthus spp* | Perennial | Phytextraxtion Phytostabilization | Zn | [30] [31] |
| *Arundo donax* | Perennial | Phytoextraction Phytostabilization | Cd, Zn, As Zn, Cr, Pb | [32] [33] [30] |
| *Zea mays* | Annual | Phytoextraction Phytofiltration | Cd, Cu, Zn, Pb Hg, Cr, Pb | [34] [35] |
| *Nicotiana tabacum* | Annual | Phytoextraction | Cu, Pb, As, Zn | [36] [37] |
| *Phytolacca americana* | Perennial | Phytoextraction Phytofiltration | Cd, Zn Cd | [38] [39] |
| *Sorghum bicolor* | Annual | Phytoextraction Phytostabilization | Cd Pb | [40] [41] |
| *Juncus acutus* | Perennial | Phytofiltration | Cr VI | [42] |
| *Fhragmites australis* | Perennial | Phytofiltration | Al, Mn, Zn, Cu, Pb, Ni, Cr, Hg | [43] |
| *Azolla caroliniana* | Annual | Phytofiltration | Hg, Cd, Pb, Cr, As, Ag, Pt, Au | [44] |
| *Lemna minor* | Perennial | Phytofiltration | Cu, Zn, Fe, Ni | [45] |
| *Gladiolus grandiflorus* | Perennial | Phytostabilization | Cu, As | [46] |
| *Vigna unguiculata* | | Phytostabilization | Pb, Zn | [47] |
| *Panicum virgatum* | | Phytodegradation | [b] PCB | [48] |
| *Mirabilis jalapa* | | Phytodegradation | [e] TPHs | [49] |
| *Eichhornia crassipex* | | Phytofiltration | Cd, Zn, Pb, Cr | [50] |

**Table 1.** *Cont.*

| Species | Life Cycle | Mechanism | Contaminants | References |
|---------|-----------|-----------|--------------|------------|
| *Festuca arundinacea* | | Phytofiltration, Phytodegradation Phytostabilization | [f] PAHs, [g] TBA, anthracene, pyrene Ni, Pb | [51] |
| *Helianthus annus* | Annual | Phytoextraction Phytostabilization Phytofiltration | As, Cu, Pb, Zn, Cd, Fe As, Cu, Pb, Zn, Hg Cd, Ni | [18] [52] [53] |
| *Salix* spp. | Perennial | Phytodegradation Phytovolatilization Phytostabilization Phytoextraction | [f] PAHs, [b] PCBs [a] TCE, [h] PCE As, Sb, Pb Cd, Zn, Cu, Pb, Ni | [54] [14] [55] [54] |

[a] trichloroethylene, [b] polychlorinated biphenyls, [c] hexachlorocyclohexane, [d] trichloroethane, [e] total petroleum hydrocarbons, [f] polycyclic aromatic hydrocarbons, [g] terbuthylazine, [h] tetrachloroethylene.

Through the phytoextraction mechanism, several species with a well-developed root system are able to absorb large quantities of pollutants, predominantly HMs, and translocate them into leaves and shoots using the solar-energy-driven transpiration pump. Therefore, toxic substances are confined and stored in metabolically inactive cellular compartments (vacuoles, cell membrane and cell wall) of the aerial parts without damaging the entire organism [56]. About 700 species of herbs, shrubs and trees, belonging mainly to *Brassicaceae*, *Euphorbiaceae*, *Asteraceae*, *Lamiaceae*, *Poaceae* and *Scophulariaceae* and including *P. vittata*, *Thlaspi caerulescens*, *B. napus*, *A. bertolonii*, *S. alfredii*, and *P. Americana* [19,26–29,38,57], are recognized as hyperaccumulators.

These plants, accumulate high level of HMs in harvestable above-ground structures [58] that can reach concentrations up to 1000-fold higher than those of common plants and range from 10,000 ppm for Zn and Mn to 1000 for Co, Cr, Ni, Pb and Cu and to 100 for Cd and Se and 10 for Hg [58,59]. Unlike hyperaccumulators, which have a low grow rate and limited biomass production, fast-growing species, such as *Z. mays*, *N. tabacum*, *H. annuus*, *S. bicolor* and *C. sativa* [18,34,36,40,41,60], can also be very effective when decontaminating polluted sites. This is due to the high amounts of produced biomass, which compensate for the low concentration of metals translocated in the aerial parts. As no additional energy is required, apart from solar energy, the possibility of recycling, the harvested biomass and the recovery of metals mainly through thermochemical methods make the application of phytoextraction methodologies sustainable and convenient at energetic and economic levels.

Several species are able to volatilize organic pollutants and HMs into the air through the leaves, epidermis, cuticle or stomata, or from the soil subsurface through root activities that, by drawing water, enhance the vadose zone thickness and the soil porosity, favoring volatile contaminant fluxes into the atmosphere [61]. The volatilization of contaminants translocated and transformed into less toxic and more volatile compounds works well as a decontamination strategy mainly for organic compounds. Volatilization is often different from transpiration because, as in the case of numerous, particularly hydrophobic, compounds, they are released into the atmosphere through the plant hydrophobic barriers, i.e., cuticle, epidermis or suberin, instead of the stomata [62]. Traditional phytoremediation plants such as willow (*Salix* sp.) and hybrid poplar (*Populus* sp.) are reported to be able to remove common groundwater contaminants, such as trichloroethylene and tetrachloroethylene [14] while alfalfa (*Medicago sativa*) is able to excrete accumulated 1,1,1-trichloroethane [61]. Moreover, phytovolatilization has also been reported for HMs such as Se, As, and Hg and has the advantage that the biomass of decontaminating plants does not require harvesting or to be adequately disposed of. For example, selenium removal is performed well by the *Brassicaceae* that, after having assimilated this metal in the organic form of seleno-aminoacids (Se-cysteine or Se-methionine), transforms them into the methylated volatile form, dimethylselenide, which is then released into the atmosphere [20]. Despite the beneficial effects of phytovolatilization, given the high dilution of contaminants in a

large volume of air and their photochemical degradation, risk assessment must also be carried out in urban areas due to the poor air quality.

Several aquatic and terrestrial plants offer the opportunity of xenobiotic removal from ground or surface water through an efficient phytofiltration mechanism that exploits the root system's ability to adsorb or absorb contaminants. Through the secretion of exudates, plant roots are able to induce rhizosphere pH modification, leading to HM precipitation on the root cell surfaces, thus preventing their leaching into the underground compartment. The absorption of harmful compounds can be followed by the precipitation inside the root cells and detoxification through chelation or oxido-reductive modifications of the metalic valence. In this regard, Dimitroula et al. [42] used the Zn-tolerant halophyte *Juncus acutus* for Cr VI removal from groundwater and the potential of endophytic strains of *Pseudomonas* and *Ochrobacterium* for detoxifying Cr VI by its reduction to Cr III. Fast-growing plants with a dense root system, such as *Azolla caroliniana*, (accumulator of Cr, Ni, Au, As, Cd, Cu, Zn, Pb, Sr and sulpha drugs), *Eichhornia crassipex* (Cd, Zn and Pb accumulator), *Lemna minor* (Zn, Cu, Pb, Fe, and Ni accumulator), and *P. australis* (Al, Mn, Zn, Cu, Pb, Ni, Cr and Hg accumulator), have provided the most promising results in the phytoremedation of aquatic systems [43–45,63]. Moreover, among common terrestrial plants, which have a longer and hairier root system than aquatic plants, several species are considered effective roothyperaccumulators, such as *Z. mays*, *A. donax*, *Triticum aestivum*, *Populus nigra*, *H. annuus*, *B. juncea* and *napus* and *P. americana* [21,22,32,33,35,39].

Other species immobilize toxic compounds at the soil–root interface without translocation into plant tissues (phytostabilization) [64]. Fast-growing plants, with deep and extensive fibrous roots, are able to limit the spread of contaminants through precipitation in an insoluble form in the rhizosphere (soil surrounding plant roots) or adsorption on lignin of the root cell walls. HMs and some organic contaminants can be converted into non-toxic form through conjugation with root exudates or reduction of metal valence [65]. Thus, by reducing the dispersion of toxic substances in the soils, their leaching into aquifers and entry into food chains is minimized. Herbaceous plants, including species of genera *Ascolepis*, *Vigna*, *Gladiolus*, *Eupatorpium* [46,47,66], are reported as being able to phytostabilize HMs such as Cu, As, Pb, Co in soil or reduce the toxicity of compounds such as the herbicide trifluralin, which is transformed from rye grass [67]. Woody plants, due to their well-developed root system that extends across a large volume of soil, are particularly suitable for phytostabilization interventions. For example, willow species are widely studied for their remediation potential [55], Eucalypts trees have been reported for their application in limiting the mobility of metals such as Cd, As, Pb, while *Ulmus glabrata* has been similarly reported to limit the mobility of As, Cu, Cr, Ni and Pb [24]. The root system also stimulates the degradation of pollutants (phytostimulation) in the rhizosphere by microbiota (bacteria, archaea and fungi) through the secretion of exudates (mainly amino acids, sugars, phenolics, organic acids), providing the carbon pool for microorganism proliferation and metabolic activities. Plant species, in turn, benefit from the detoxification of microorganisms, which reduces stress on plant growth. The presence of plant roots also creates microenvironments that promote microbial growth (the rhizosphere population can be several orders of magnitude higher than non-vegetated soils) and microbial remediation action, favoring soil aeration and water drawing and enhancing the bioavailability of contaminants, through their detachment from soil particles [17,68]. Rhizosphere macro-fauna (nematodes, protists, collembola, and earthworms) also strengthen root–microbe interactions, potentiating rhizoremediation activity [69]. Organic contaminants can also be biodegraded up to mineralization or transformed into smaller molecules that are less toxic or non-toxic by extracellular enzymes such as laccases, oxidases, dehalogenates and nitrore-ductases, secreted by the root system. Moreover, xenobiotics undergo transformation into non-hazardous components in the above-ground parts of plants (phytodegradation), by being metabolized and becoming part of vegetal tissues. For example, PCBs can be detoxified through dechlorination by species such as *Ipomea balsamina*, *Mirabilis jalapa*, *P. virgatum* and *B. napus* [48,49,70], while the hexachlorocyclohexane (HCH) can be similarly detoxified

via appropriately selected *Populus* clones [17]. Hannink et al. [71] studied the effect of the nitroreductase enzyme produced by the roots of *N. tabacum*, which contributes to the degradative action on trinitrotoluene. Petroleum, containing saturated aliphatic alkanes and PAHs, is a carbon and energy source for the rhizosphere microbiota, which removes these compounds by mineralization into $H_2O$ and $CO_2$ [72]. Among soil microorganisms, bacteria are responsible for the degradation of contaminants, and Proteobacteria genera such as *Burkholderia*, *Alteromonas*, *Pseudomonas*, *Rhodococcus*, *Xanthomonas*, *Caulobacter* together with the *Actinomycetes* group contain the main species involved in the pollutants rhizodegradation [73]. Several plants harboring microbial hydrocarbon degraders, such as *H. annus*, *Z. mais*, *Festuca arundinacea*, *M. sativa* and *Salix viminalis*, are responsible for the decontamination of highly polluted sites [51,54,74]. Further, poplars are the most commonly used species in phytoremediation projects, as they are perennial, fast-growing trees with a broad adaptability to different climatic conditions. Their expansive deep root systems, typically spreading up to two or three times the height of the tree, grow near aquifers, drawing huge water amounts and dissolving contaminants. Interestingly, poplars are among the few tree species that are able to establish a mutualistic symbiotic association between the root system and mycorrhizal fungi by expanding the root surface up to 800-fold. This also largely favors the rhizodegradation capacity of poplar trees by stimulating the degradation activities of rhizosphere bacteria. Finally, the enormous intraspecific genetic diversity is the basis of the possibility of finding proper poplar clones endowed with suitable characteristics for the detoxification of various xenobiotics in different environmental conditions. Poplar phytoremediation applications range from sites contaminated by hydrocarbons [75] to those polluted by chlorinated organic compounds. They can degrade trichloroethylene (TCE) into trichloroethanol, di and trichloroacetic acid, or $CO_2$ and $H_2O$ [14]. They can also uptake atrazine and degrade it into less toxic compounds [15]. Poplars can also adsorb PCBs through their roots, translocate them into their shoots and carry out their degradation [76]. Moreover, Bianconi et al. [17] have demonstrated that, by combining the use of suitable poplar clones and soil inoculation with properly degraded bacterial strains, HCH concentration in contaminated areas can be significantly reduced.

## 3. Valorization of Phytoremediation Byproducts

The integration of the produced biomass in eco-sustainable processes for its valorization increases the economic and environmental benefits of phytoremediation, but has suffered several drawbacks. Despite numerous plants having demonstrated a strong HM extraction ability, their slow growth rate, due to the difficult adaptation to the drastic conditions of contaminated soils, and the long times required for detoxification (2–60 years for hyperaccumulators and 25–2800 for non-hyperaccumutators), can reduce the efficiency of remediation interventions. Furthermore, the root system length is often not sufficient to reach the entire depth of contaminated areas. This factor, combined with the dependance of growth rate on seasonality, weather conditions, plant diseases, and pests limits the success of decontamination [58]. The introduction of foreign natural or transgenic species, provided with suitable remediation capacity in polluted sites, especially if they can show characteristics of infestation, represents a threat to the structure and equilibrium of the local vegetal community and of the entire ecosystem. The risks of a drive towards a more homogeneous flora, altering the natural biodiversity, or to the transfer of genes to the environment, must be considered. Careful attention must also be paid to the introduction of cultivation methods or to the utilization of chemical compounds that modify soil structure or the mobility and availability of contaminants. The alteration of the structure of the soil and of the autochthonous microbial communities and the possible leaking of contaminants in other compartments, such as aquifers, represent serious problems to be avoided. Finally, among the most important drawbacks of phytoremediation is the safe disposal and utilization of the resulting biomass, which requires thorough caution to avoid the serious risks linked to the environment and to trophic chain-related secondary pollution. The chemical form of HMs and the plant characteristics must be considered in order

to predict the fate of these contaminants when planning biomass disposal solutions in the different streams of bio-waste utilization processes. The potential risks of secondary pollution and contaminant incorporation into trophic chains can be reduced through a properly programmed harvest of biomass, followed by the recovery of toxic compounds and the exploitation of plant residues. In this regard, the main concerns originate from HMs, which are non-degradable toxic elements, different from organic pollutants, and are bioaccumulated into trophic chains. To date, about 700 species of hyperaccumulator plants are known, these show the ability to extract huge quantities of metals and metalloids and store them in vegetal tissues in quantities hundreds or thousands of times higher than in normal plants [77]. The exploitation of HM-containing biomass, following a biorefinery approach and with zero waste production, strengthens the phytoremediation status of an economically advantageous and eco-sustainable process. Adequate solutions for lignocellulosic bio-waste valorization involve the use of the organic fraction (lignin, cellulose, and hemicellulose) and the properly managed recovery of high-value metallic elements that represent attractive secondary resources, as shown in Figure 2.

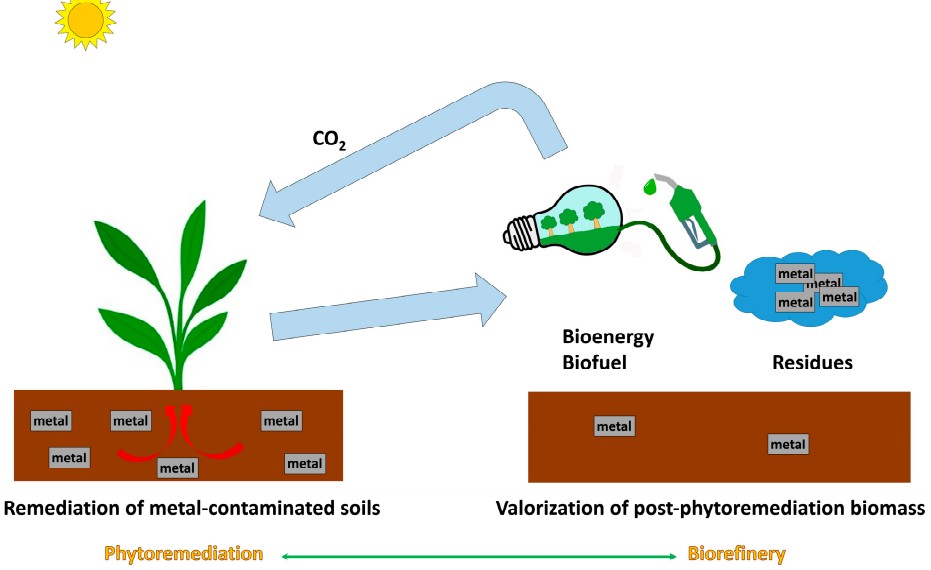

**Figure 2.** Schematic representation of "Sustainable phytoremediation" for decontamination of polluted lands decontamination and biofuel production through a biorefinery approach toward a zero-carbon-emission circular bio-economy.

The concept of the re-utilization of biomass, a key component of central value chains, embraces one of the most effective approaches by which the circular economy aims to restore HM polluted sites and replace the dependency on fossil fuels with a zero-carbon-emissions bioeconomy, and is included in the strategic program for the mitigation of greenhouse warming effects. This "sustainable phytoremediation" approach demonstrates its higher potential compared with chemical techniques, as it represents an environmentally friendly solution to pervasive pollution issues, and points directly to a new model of inclusive social growth of the economy based on the efficient and zero-waste exploitation of byproducts for sustainable development [78].

### 3.1. Thermochemical Methods

In recent decades, two alternative strategies for the valorization of post-phytoremediation residues involve the thermochemical or biochemical conversion of biomass for bioenergy and the recovery of value-added products.

The accumulator and hyperaccumulator plants directly cause one of the major problems related to the phytoremediation technique, as the process itself involves the bioaccumulation of contaminants within the green biomass [78]. Consequently, the

collection and treatment of such biomass can generate toxic waste and cause secondary pollution, if improperly managed. The most widely used methodologies for treating plants from phytoremediation and, at the same time, produce thermal energy, are thermochemical processes.

These methodologies (pyrolysis, incineration, and gasification) transform the organic fraction into solid (biochar, ashes), liquid (bio-oil) and gaseous (syngas, $H_2$, $CO_2$ and CO) products, exploitable for heat, electricity generation at large scales and biofuel production, and allow the recovery of valuable metallic elements through their concentration followed by extraction or encapsulation into the solid fraction [79]. The main drawbacks of thermochemical treatments consist in the difficulty to follow the HM partition among the different final products, leading to potential metal reintroduction into the environment. Critical considerations have highlighted that, instead of removing HMs from biowastes, their immobilization can efficiently limit discharge into the environment. Pyrolysis is one of the most suitable methods for HM stabilization such that, through proper operating conditions related to temperature, time, and catalyst utilization (metal oxides, dolomite, zeolites), can be concentrated and encapsulated into a solid residue (biochar). This is of fundamental importance when efficiently limiting metal mobility and bioavailability in ecosystems as well as obtaining gas and bio-oil byproducts with low or no presence of contaminating HMs. Moreover, metals containing biochar find valuable application in pollution-resolving issues such as dye removal or the elimination of organic contaminants from wastewater, through photocatalytic activity (Cd biochar) [80,81]. This is also the case for Mn-containing biochar, obtained from *Phytolacca acinosa* Roxb., and post-phytoremediation residues that have been utilized for the combined decontamination of Pb and tetracylin, as reported by Zhou et al. [82].

In addition, hydrothermal gasification, liquefaction and carbonization are also used, though these need the presence of subcritical or supercritical water [83].

The different techniques in practical application should be based on bio-waste from hyperaccumulator species, because HMs from the enriched biomass could be released during the thermal treatment processes [84]. Furthermore, the presence of HMs can damage thermal conversion systems, with possible product degradation [85], and, from an economic point of view, the combustion of biomass from phytoremediation does not allow the total recovery of the energy stored in the substrates [86].

Metallurgy is the method generally carried out for metals recovery from plants used in phytoremediation. This extractive technique is able to remove heavy or precious metals from the biomass, refining the extracted raw metals into a purer form [87]. In addition, pyrometallurgy is widely used for metals that are leached, then recovered, from the ash of incinerated biomass. Additionally, hydrometallurgy, a method using compressed hot water, involves the leaching of metals directly from the substrate [88].

### 3.2. Biological Methods

Although thermochemical treatments have proved to be promising strategies, the use of biocatalysts (enzymes and microorganisms) represents a powerful alternative. Recently, the utilization of post-phytoremediation residues has attracted increasing attention as one of the best sustainable resources for bio-ethanol production. Plants grown on marginal, non-productive lands, allow soil competition for food crops that were the previous carbon rich substrates (sugarcane, corn, wheat, sweet potato, barley) utilized for the production of first-generation bioethanol to be overcome. Consequently, in second-generation bioethanol, high lignocellulose-containing residues from the agro-industrial sector (corn stover or cobs, brewery spent grains, wheat straw, sugarcane or beet molasses among others), are the preferred feedstocks, due to the absence of competition with food sources [89]. In this context bio-wastes from phytoremediation practices represent attractive substrates due to the lack of commercial value because their utilization offers the opportunity to address, in an economically viable way, disposal concerns of HM-contaminated residues. The presence of high concentrations of toxic HMs in hyperaccumulator plants is the real challenge for the

safe and profitable exploitation of biomass in biocatalyst-mediated transformations. The yield of bioethanol production processes, which consist of sequential enzymatic hydrolysis for the release of soluble sugars and their fermentation, can be affected by the inhibitory action of HMs on both enzymatic and microbial catalysts. For the different plant species, the knowledge of metal concentrations in the biomass and the distribution of HMs in the different vegetal tissues are of high importance when choosing adequate biocatalysts and determining the conditions for maximizing saccharification and fermentation yields [90].

The problem of the presence of toxic metal is particularly significant for biogas production from post phytoremediation residues through the anaerobic digestion process, which is among the most efficient technologies for renewable energy production from lignocellulosic residues [91]. The process consists in the conversion of the organic multi molecular compounds (proteins, polysaccharides, lipids), into simpler intermediates (soluble sugars, amino acids), which are further transformed through gasification into $CO_2$ and $CH_4$ by the cooperation of the anaerobic and relatively anaerobic bacterial populations of the fermentative, syntrophic, acetogenic, and methanogenic microorganism groups [92]. Several authors have demonstrated a sensitivity to the presence of metal in anaerobic microbial communities, which can be positively or negatively affected by contaminations of HMs, depending on their nature, concentration and specific process features [93,94]. Different studies have reported the ranges of the inhibitory heavy metal concentrations in anaerobic digestion processes. Some of these are represented by the following values: >20 mg/L for $Cd^{2+}$, >1 mg/L for $Cu^{2+}$, >10 mg/L for $Ni^{2+}$, and >4 mg/L for $Zn^{2+}$ [95]. In particular, the high vulnerability of methanogens is highlighted, and Cu and Zn are the metals that most affect these microorganisms [96].

Knowledge of the destinations of metal into the different vegetal tissues is crucial for the evaluation of the most suitable strategies for bio-waste exploitation toward bioenergy. For instance, the exploitation of plant parts, excluded from metal accumulation or that receive metal amounts within certain threshold values, is a feasible alternative to expensive pretreatments. In this regard, due to the limited translocation of HMs from roots to shoots of *H. annuus*, its aboveground biomass was effectively used in long-term stable anaerobic digestion processes [97]. In a continuous stirred tank reactor, the measured soluble metal concentrations were found to be far below the inhibitory levels allowing stable process operativity for 20 days. Moreover, the microbial anaerobic communities did not suffer the presence of toxic metal, maintaining an adequate structure and diversity that is effective for the lignocellulosic substrate degradation [95]. The deep understanding of the metal's fate inside the plants is also indispensable in order to address the concern of the possible release of the toxic compounds back into the environment. The encountered lack of social acceptance toward the exploitation of post-phytoremediation residues for bioenergy purposes can be overcome by the implementation of legislative systems that, based on unbiased scientific knowledges, can adequately regulate the sustainable valorization of phytoremediation crops, with the environment and human health as central issues [98]. The phytoremediation strategies that allow the effective and sustainable management of contaminated sites and of biomass exploitation toward bioenergy require the careful consideration of different variables. The remediation potential of the different species, the characteristics of polluted environments, the distribution of metals in the various streams during the bioenergy production processes are the main evaluations required. An ideal and predictable way of concentrating the metal fraction during biomass transformation should be identified so that the partition into different products can be clearly determined. In fact, only a low metal content or its absence determines the exploitability of the different products (liquid biofuels such as bioethanol or biodiesel, biogas and bio-fortified dietary supplements enriched with Se, Fe, or Zn) without concerns for human and environmental health. However, since it has been clearly demonstrated that the fate of metals is regulated by multifactorial variables, case-by-case evaluations are necessary for adequate phyto-management [99]. Additionally, rigorous risk assessments are required for the choice of post-phytoremediation crops for bioenergy purposes and for the drive toward plants that

utilize the phytostabilization mechanism. These species, unlike the "hyperaccumulators", characterized by high concentrations of metals in upper-ground plant tissues [77], tolerate the presence of metals at elevated amounts in the soils and stabilize the contaminants in the rhizosphere by reducing their bioavailability, hindering their migration due to wind and water erosion, and by leaching into the groundwater. Different studies propose strategies that combine phytostabilization interventions on metal-contaminated soils with the production of bioenergy crops in order to maximize the benefits of phytoremediation.

Despite the critical issues discussed, the "sustainable phytoremediation" strategy addresses well the challenge of using polluted marginal soils for the cultivation of promising dedicated energy crops in order to ensure the restoration of contaminated lands while providing valuable biomass for bioenergy purposes. Efforts by scientists, policy makers and corporations in several countries are focused on identifying and exploiting promising high-biomass energy crops towards the goal of a biofuel production that will account for over a quarter of global demand for transport fuels, such as those imposed by the International Energy Agency for 2050 [100]. Their attention has been focused mainly on non-edible perennial species with properties suitable for soil phytoremediation and excellent energetic attributes (no need of harsh pretreatments due to low lignin content and the presence of a high cellulosic carbohydrates fraction). Though potential for biofuel production is mainly explored among fast-growing perennial woody species, usable in short rotation coppice programs (poplar, willow, eucalyptus, black locust) [101], several perennial herbaceous species have gained increasing attention. These combine a high phytoremediation capacity and important second-generation features (non-edibility, ability to live luxuriously in marginal soils and in harsh environmental conditions such as drought, elevated salinity, and the presence of toxic compounds) with high value for bioenergy purposes. Several perennial plants, including *A. donax* (giant reed), *Miscanthus* spp, *P. virgatum* (switch grass), *H. annuus* (sunflower), and *P. australis* are able to remove HMs and persistent organic pollutants. Among these, *A. donax* and *Miscanthus* spp. are recognized as robust energy crops with a low requirement for fertilizers and pesticides due to their low susceptibility to diseases and their ability to thrive in different unfavorable conditions, such as contaminated ecosystems. *A. donax*, together with *P. virgatus*, shows one the highest biomass yields of around 20 tons/ha/year, while *Miscanthus* spp. accumulates HMs prevalently in the roots and rhizome, and degrade organic pollutants, such as PAHs, hydrocarbons and pesticides, through root exudates [31]. These are just a small number of examples among the numerous plants whose exploitation can provide valuable solutions by which to address the two current most urgent global challenges represented by general pollution issues that impact the health of the entire biota and the growing demand for bioenergy. In this direction, future efforts will still be needed to identify the most suitable energy crops for combined phytoremediation and bioenergy production processes with favorable life cycle assessment outputs [102].

## 4. Pretreatment Technology on Biomass from Phytoremediation

To achieve a high-performance process, bio-accumulated metals and other pollutants should be removed and recovered without degrading biofuels and bioenergy production [103]. The mentioned thermochemical methodologies, although widely used, do not take into account the possibility of using biomass as a source of biofuels or as renewable biochemical substances [104].

Within the various technologies for the recovery and use of biomass from phytoremediation, a series of limitations on their use must also be taken into account. The literature indicates that the concentration of contamination, toxicity and bioavailability, choice of plants and stress tolerance are the main aspects to take into account. The drawbacks of phytoremediation are the accumulation of pollutants in the edible parts of the plants, specific accumulation of a single metal, and the slow speed of the method, due to the need for numerous subsequent planting cycles for decontamination. The treatment of plant biomass after phytoremediation may cause additional environmental pollution due to the

release of absorbed contaminants, the transfer of HMs to other matrices (air and water) and their absorption into the food chain, potentially risking human and animal health [3]. The disposal and re-use of contaminated biomass is extremely important. Many techniques, such as biodegradation, pyrolysis and incineration, present a high risk of secondary pollution. An effective strategy will be the combination of these different treatments, which can overcome the disadvantages of polluted biomass use. Therefore, it is important to explore the extraction methods of different forms of HMs in plants and to evaluate the possibility of the leaching of HMs from plants in different applications, and to then carry out appropriate disposal of the contaminated biomass [79].

Biochemical processes involving the use of appropriate hydrolytic enzymes and/or whole microorganisms, are practical non-polluting alternatives for the conversion of plant biomass into renewable resources [105]. However, due to its complex structure, consisting of carbohydrate polymers mainly composed of cellulose (40–50%), hemicelluloses (20–30%) and lignin (10–25%) [106], lignocellulosic biomass from phytoremediation requires pretreatment in order to access the different components of the bioenergy feedstock [107]. Lignin, with its binding function, imparts rigidity to plants thanks to the lignin–carbohydrate complex, and causes the inhibition of the enzymatic hydrolysis process of structural polysaccharides. Therefore, biomass pretreatment should aim, after cost reduction and the use of processes with low environmental impact, to avoid complete carbohydrate degradation, improve sugar yield after enzymatic digestion and prevent the production of compounds inhibiting the hydrolysis and fermentation processes [108].

There are several widely used pretreatment techniques that improve the accessibility of lignocellulosic biomass to enzymes by implementing the saccharification process, which in turn can be divided into physical, chemical, and biological methods [108].

Based on bioethanol research and high value-added production, HM-contaminated feedstocks are required for a pre-processing step, useful for metal extraction and limiting inhibition in the hydrolytic and fermentation steps [109]. The removal of HMs and other pollutants from biomass depends on the method of sequestration of pollutants adopted by the different species used for phytoremediation [79].

Traditional physical techniques, such as millers, grinders, and ultraviolet or microwave radiation are aimed at reducing the size and crystallinity of lignocellulosic biomass. Physical methods are usually used in combination with other pretreatments, allowing one to disrupt the internal bond between lignocellulosic biomass with little carbohydrate loss [110]. Chemical and physical–chemical methods are largely adopted for the lignocellulos's breakdown, even if the production of inhibitors can be a common drawback [111].

An example of physical treatment is steam explosion (SE), reported by Ziegler-Devin et al. [112], which achieved the decontamination of trace elements (Zn, Mn) combined with the deconstruction of woody biomass. The willow obtained from phytomanaged plots of rotational coppice forest, harvested on contaminated soil, was treated with 2% sulfuric acid followed by an increase in temperature (220 °C), reaching 80% extraction of Mn and Zn in the water effluent.

Liquid extraction is a widely used treatment by which to transfer metals from the solid to the liquid phase [113]. *P. vittata*, as a hyperaccumulator applied to the phytoremediation of contaminated sites [25], has been successfully pretreated using diluted acidic (1% $HNO_3$) and alkali (1% NaOH), which allows the removal of more metal than does the use of ultrapure water [105]. In particular, the best saccharification yields were obtained by treating the phytomass with diluted acid under stirring, which is also suggested as an economical method for Arsenic extraction to convert the phytomass into bioethanol. Another effective method to recover up to 93% of As from *P. vittata*, is coupling ethanol extraction with anaerobic digestion and As–Mg precipitation of digestate supernatant [114].

Asad et al. [115] investigated the pretreatment of non-woody lignocellulosic (tobacco) and woody biomasses enriched with trace elements, and the fractionation of the biomass was also described.

Three processes, ethanol organosolv, dilute acid and alkaline pretreatments, widely used for cellulosic bioethanol production [116] were evaluated. The results indicate that the best performing pretreatments were conducted under acidic conditions at 2% $w/w$ sulfuric acid, at a temperature of 170 °C, achieving up to 90% of the metals (for Zn and Mn) in the effluent water, obtaining a clean substrate. The alkaline conditions, using 15% $w/w$ of NaOH (170 °C), resulted in a lower extraction of metals, mainly concentrated in the cellulosic pulp (70–98%). The pretreatment that returned lower extraction yields was with organosolv, where the metals were mainly found in the pulp, in the effluent water and in the lignin.

Three chemical pretreatments were also performed on two rapeseed cultivars [117], an important oil crop, grown in the presence of $CdCl_2$. One-step chemical pretreatments with alkali (NaOH at different concentrations) extracted only 34% and 58% Cd in the analyzed rapeseed, with better Cd extraction using one-step acidic extraction (with $H_2SO_4$). The optimal pretreatments were two-step chemical methods that allowed the complete Cd release (99%) in the two mature stalks, simultaneously resulting in an enhancement of biomass saccharification.

Recently, diluted sulfuric acid treatment was applied on sweet sorghum bagasse biomass, grown on Cadmium-contaminated soil [118]. The pretreatment allowed the Cd to be completely released and the accessibility of the biomass to subsequently be improved by enzymatic hydrolysis, with the hemicelluloses fraction almost completely degraded and lignocellulosic structures deconstructed.

The use of deep eutectic solvent (DES) was explored by Zhang et al. [119]. DES was able to chelate HMs in modified poplar and the hyperaccumulator *S. alfredii*, achieving 98.3% Cd and 94% Cu extraction, along with a clean cellulose-rich substrate.

*Miscanthus* bioenergy feedstock, used for phytoremediation of soil rich in HM contaminants, was treated with an ionic liquid [120]. The ionoSolv process conducted on the biomass (1 h at 150 °C,) allowed the obtainment of a clean pulp rich in cellulose, with the removal of hemicelluloses and 60% of lignin. Furthermore, the described methods also caused the electrodeposition of metals, resulting in total Pb extraction (99.3%) and a recovery of between 96 and 98% of Cd, Cu and Zn, thanks to the action of the ionic liquid as a metal dissolving agent [121].

Recent research on poplar residues, collected in southern Italy (Apulia region) and grown on soil contaminated by HMs and PCBs [16] was chemically pretreated using formic acid and hydrogen peroxide at different concentrations. With the treatment using performic acid at 7 M, the complete removal and dissolution of lignin (100%) and xylan was achieved, obtaining a clean cellulose pulp. A pretreatment carried out under mild conditions (3.5 M performic acid at 55 °C for 4 h), allowed good results in the removal of lignin (>75%). Moreover, most of the minerals present in the resulting poplar biomass dissolved in the aqueous phase [122].

Another example of physico-chemical treatment was carried out on a napier grass biomass, obtained after phytoremediation of soils polluted by HMs, which was steam exploded (180 °C, 10 min), after an immersion of 24 h of the feedstock in 1.5% $H_2SO_2$ [90].

A biocompatible pretreatment for the use of lignocellulosic biomass is a biological pretreatment. The biological technique is characterized by low cost and simplicity of operation, is environmentally friendly and does not lead to the formation of inhibitors during the process [107]. The method by which the biological pretreatment occurs is with the use of fungi and bacteria capable of making the lignin component more easily attackable [123]. Fungi are usually used in biological pretreatment because they can produce enzymes that can effectively degrade lignin and hemicelluloses [124].

In addition, to make the lignocellulosic biomass more hydrolysable, some microbial and bacterial consortia, as well as enzyme cocktails consisting of lignin peroxidase, Mn peroxidase and laccase, are also used to deconstruct lignocellulosic biomass [125]. Although more biocompatible, the biological approach is usually slower and has a lower efficiency than other chemical or chemical–physical pretreatments, especially for industrial purposes.

Moreover, it is not always possible to separate HMs from the biologically treated plant and there could be the problem of the formation of leachate and residue that still contains high concentrations of metals. This method can be taken into consideration when the HM content in phytoremediation plants is low.

Waghmare et al. [126] investigated the effectiveness of a plant–bacterial consortia, using *Pogonatherum crinitum* and *Bacillus pumilus* strain PgJ, for textile effluent phytoremediation. This technique has made it possible to reduce the toxicity load in the effluents and to improve the germination of *Phaseolus mungo* and *Sorghum vulgare* seeds [126].

A fungal pretreatment was implemented on sweet sorghum bagasse (SSB), a biomass that has received great attention for its use in phytoremediation and biofuel production [40]. *Coriolus versicolor* in bioreactor increased the production of several laccase and xylanase, resulting in high lignin degradation [124].

Table 2 details the above-mentioned pretreatment processes on polluted biomass from phytoremediation.

**Table 2.** Pretreatments on biomass from phytoremediation, advantages and disadvantages.

| Pretreatment | | Biomass | Advantages | Disadvantages | Ref |
|---|---|---|---|---|---|
| Physico-chemical | Steam explosion–sulfuric acid | Willow, Napier grass | Cellulose enrichment. Lignin transformation. High rate of metals removal | High operating temperature. Generation of toxic compounds. | [112] |
| | Nitric acid | *P. vittata* | Lignin solubilization. Cellulose crystallinity reduction | Cost associated with acids and recovery | [105] |
| | Sulfuric acid | *N. tabacum* L., *S. viminalis*, *B. pendula*, *B. juncea* L., Sweet sorghum bagasse | Efficient extraction of the metals (80% As, up to 90% Cd). Glucan enrichment. | Hemicelluloses degradation. Formation of inhibitors, lignin breakdown products. Cost associated with acids and recovery | [115,117,118] |
| | Sodium hydroxide | *N. tabacum* L., *S. viminalis*, *B. pendula*, *P. vittata*, *B. juncea* L. | Lignin removal (up to 80%). Easy sugar recovery. | Low metal extraction. Expensive | [105,115,117] |
| Chemical | Sodium hydroxide + Sulfuric acid | *B. juncea* L. | Complete metal (Cd) release (99%). Cellulose crystallinity reduction | High costs | [117] |
| | Ethanol extraction | *P. vittata* | Low soluble carbon reduction. Efficient Metals extraction (As 93%). | Expensive | [114] |
| | Ethanol organosolv | *N. tabacum* L., *S. viminalis*, *B. pendula*, | High rate of lignin solubilization. | Metals extraction is low. High production costs. | [115] |
| | Deep eutectic solvent (DES) | *S. alfredii* | Lignin (90%) and hemicellulose removal. Cellulose enrichment. | Low cellulose-rich pulp obtainment. High viscosity at room temperature. Toxicity. | [118] |
| | IonoSolv | *Miscanthus* | Biomass enriched in cellulose. Lignin removal. Effective extraction of HMs | Degradation of hemicelluloses. Costly solvents. | [121] |
| | Organosolv (formic acid + hydrogen peroxide) | Poplar | Complete removal of lignin. Obtainment of a clean cellulose pulp. Dissolution of metals | Xylan removal. Not be applied to softwoods. External energy requirement. | [122] |
| Biological | *C. versicolor* | Sweet sorghum bagasse | Low cost. Environmentally friendly. No formation of inhibitors | Not usable with high HM content. Long treatment. Low hydrolysis rate. | [124] |

## 5. Biofuels and High Added Value Production

The biomass produced during phytoremediation can be exploited for its added value. The valorization of biomass resulting from phytoremediation, with the production of bioethanol, biodiesel, biogas, and biochemical products (Figure 3), may not only help to meet global energy demand, but also provide a boost towards the transition to a circular and sustainable economy [127].

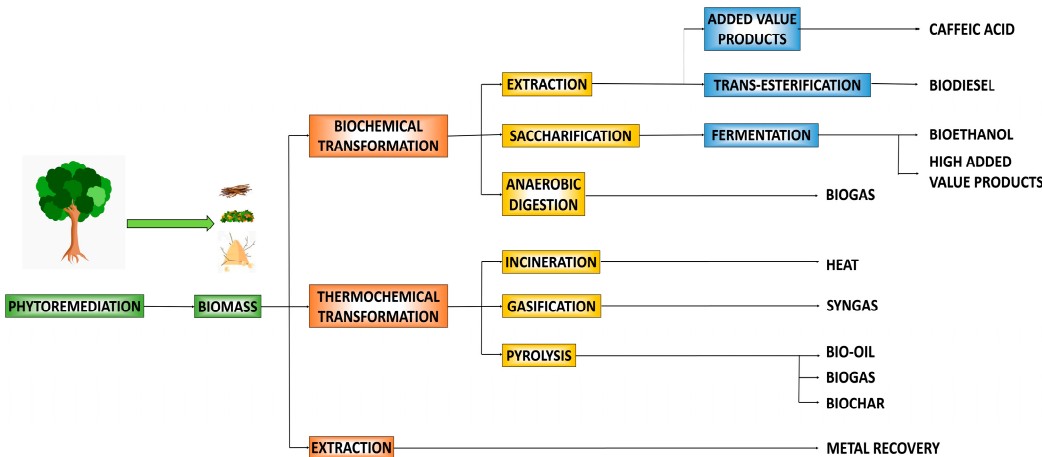

**Figure 3.** Schematic illustration of different routes for post-phytoremediation biomass reutilization towards biofuels (bioethanol, biogas, biodiesel), added value products (biochar, bio-oil, caffeic acid) and metal recovery.

To date, numerous studies report the importance of the ability to couple the potential of phytoremediation with bioenergy implementation.

A conversion process, necessary to transform the biomass into the desired product, follows the pretreatment process. Specifically, the following sections will examine the production of bioethanol, biogas, and biodiesel from phytoremediation plants (Table 3) using biochemical methodologies. The most common and environmentally friendly biochemical conversion methods involve hydrolysis, enzymatic catalysis and fermentation of the biomass.

**Table 3.** Feedstock from phytoremediation to produce biofuels and high value products.

| Feedstock | Pretreatment | Metal Detected | Product Target | References |
|---|---|---|---|---|
| *Avena sativa* L. | Mechanical treatment Anaerobic digestion | Cd | Biogas | [128] |
| *Betula pendula* | 2% $H_2SO_4$ 15% NaOH Ethanol organosolv | Zn, Mn Trace elements Trace elements | Bioethanol | [115] |
| *Brassica juncea* L. | 1.0%, 2.0%, 4.0%, 8.0% NaOH 2.0%, 4.0%, 8.0%, 12%, 16% $H_2SO_4$ 4.0% NaOH + 2.0%, 4.0%, 8.0% $H_2SO_4$ | Cd | Bioethanol | [117] |
| *Brassica napus* | Mechanical treatment Anaerobic digestion | Cd, Cu | Biogas | [129] |
| *Eichhornia crassipes* | 1% NaOH—3% $H_2SO_4$ Mechanical milling | Cu, Cr | Bioethanol Biohydrogen Biogas | [50] [130] |
| *Elsholtziahaichowensis* | Anaerobic digestion | Cu, Pb, Zn, Cd, Mn, As | Biogas | [131] |

**Table 3.** *Cont.*

| Feedstock | Pretreatment | Metal Detected | Product Target | References |
|---|---|---|---|---|
| *Elsholtzia splendens Nakai* | Anaerobic digestion | Cu | Biogas | [129] |
| *Helianthus annuus* | 2% NaOH at 50 °C<br>*Phanerochaete chrysosporium*<br>Anaerobic digestion<br>Aerobic digestion | Ni, As, Pb, Cu, Cd, Zn<br>Ni<br>Trace elements | Bioethanol<br>Value-added products<br>Biogas<br>Compost | [132]<br>[133]<br>[52]<br>[53]<br>[95] |
| *Lepidium sativum* L. | Anaerobic digestion | Hg | Biogas | [134] |
| *Mentha spicata* | Anaerobic digestion | Hg | Biogas | [134] |
| *Miscanthus sinensis* OPM-10 | 1-methylimidazolium chloride | Pb, Zn, Fe, Cu, Cr, Ni, As, Cd | Biorefinery | [120] |
| *Nicotiana glauca* | Aerobic digestion<br>Anaerobic digestion | Trace elements | Biogas<br>Compost | [53] |
| *Nicotiana tabacum* L. | 2% $H_2SO_4$<br>15% NaOH<br>Ethanol organosolv | Zn, Mn<br>Trace elements<br>Trace elements | Bioethanol | [115] |
| *Oenothera biennis* L., | Anaerobic digestion | Cu | Biogas | [129] |
| *Oryza sativa* L. | 6% NaOH | Cd, Pb, Cu, Zn | Biogas | [135] |
| *Pennisetum purpureum* | Acid (3% $H_2SO_4$) Steam explosion | Zn, Cd, Cr | Bioethanol | [90] |
| *Phytolacca americana* L. | Anaerobic digestion | Cu, Pb, Zn, Cd, Mn, As | Biogas | [129,131] |
| *Piptatherum miliaceum* | Aerobic digestion<br>Anaerobic digestion | Trace elements | Biogas<br>Compost | [53] |
| *Pogonatherum crinitum* | *B. pumilus* | | Bioethanol | [126] |
| *Populus nigra* | 1.5–7 mol/L Performic Acid | Trace elements | Levulinic acid | [122] |
| *Pteris vittata* | ultrapure water, 1% $HNO_3$, 1% NaOH, shaking (200 rpm) and ultrasonication (40 kHz).<br>35% Ethanol—anaerobic digestion | As, Mg | Bioethanol<br>Biogas | [105,114] |
| *Salix viminalis* W | Steam explosion<br>2% $H_2SO_4$<br>15% NaOH<br>Ethanol organosolv | Trace elements<br>Zn, Mn | Bioethanol | [112]<br>[115] |
| *Sedum alfredii* | choline chloride/lactic acid (deep eutetic solvent)<br>Anaerobic digestion | Cd, Cu, Pb, Zn, Cd, Mn, As | Value-added products<br>Biogas | [119,131] |
| *Silybum marianum* | Anaerobic digestion | Trace elements | Biogas | [52,53] |
| *Sinapis alba* | *Phanerochaete chrysosporium.* | Ni | Value-added products | [133] |
| *Solanum nigrum* | Anaerobic digestion | Cu, Pb, Zn, Cd, Mn, As | Biogas | [131] |
| *Sorghum bicolor* | 0.5, 0.75, 1.0, 1.5% $H_2SO_4$<br>Mechanical milling<br>Alkaline (2% NaOH) Steam explosion<br>$HNO_3$ | Pb, Cu, Zn, Cd | Bioethanol<br>Biogas<br>Biofuels<br>Organic fertilizer | [136,137] |
| *Trapa bispinnosa* | Mechanical milling | Cu, Cr | Biogas | [130] |
| *Zea mays* | Anaerobic digestion<br>*Rhizophagus irregularis* and *Cupriavidus* sp. | Cu, Cd, Zn | Biogas | [129,138,139] |

*5.1. Bioethanol*

First-generation bioethanol, produced by the fermentation of glucose from sugar- or starch-rich biomass, such as beetroot and corn, presents the problem of using food for fuel production. The production of second-generation bioethanol, which exploits lignocellulosic biomass, including that resulting from phytoremediation, represents an environmentally sustainable compromise. Such biofuels are referred to as zero-carbon products, thus reducing greenhouse gas emissions [140]. Furthermore, the versatility of these biomasses used for phytoremediation has increased the added value for the biofuels production.

The available literature regarding the production of bioethanol from biomass resulting from phytoremediation is very low. Although it has been necessary to consider on a case-by-case basis the level of soil pollution and the method used to carry out the phytoremediation, the published studies clearly indicate that the phytomass can be managed in a sustainable way, allowing its conversion into bioethanol and bio-based products. Tusher et al. [105] have developed a bioprocessing system by which to obtain bioethanol from *P. vittata*. The chemical pretreated phytomass was saccharified by using a recombinant yeast *Kluyveromyces marxianus* (KR7 strain) and two recombinant *Bacillus subtilis* strains. A one-step method, involving the six cellulases from KR7, has highlighted the capability to convert *P. vittata* into reducing sugars to a greater extent than *Bacillus* strains. Instead, the co-culture strategy with the use of KR7 and one of the recombinants *B. subtilis* Type 2 (containing cellulosomal genes) enabled better results for bioethanol production, reaching 48.5% ethanol yield. In addition, the feasibility and affordability of bioethanol production represent a crucial factor in the use of energy crops.

Bioethanol production was explored on two *B. juncea* L. cultivars, grown in the presence of Cadmium [117]. The biomass pretreated by one-step (acid or alkaline) and two-step methods (NaOH and $H_2SO_4$) showed an improvement in saccharification yield, conducted using a commercial enzyme cocktail containing cellulase, β-glucanaseand xylanase, along with complete metal release.

The subsequent fermentation process, conducted using *Saccharomyces cerevisiae*, confirmed the greater yield of bioethanol, which increased by 8% and 12%, respectively, for the two cultivars, using the two Cd-accumulated stalks, when compared with the control. The utilized methodologies emphasized the integration of biofuel production with the environmental recovery of the territory and the production of a minimum amount of waste.

Napier grass, which had a high concentration of Cd, Cr and Zn after phytoremediation [90] and upon steam explosion treatment, was subjected to enzymatic hydrolysis with a mixture of cellulolytic and xylanolytic components. The presence of Zn and Cd decreased glucose recovery after enzymatic action, particularly for lower enzyme dosages, when compared with the control. In contrast, the presence of Cr improved glucose recovery at all concentrations tested. In addition, ethanol production, after simultaneous fermentation and saccharification with *E. coli*, was positively affected in the presence of the tested HMs. A stepwise process by which to produce bioethanol from *P. crinitum*, grown in polluted textile wastewater, was conducted by Waghmare et al. [126]. Sugar production using hydrolytic enzymes of *Phanerochaete chrysosporium*, achieved a good yield of reducing sugars. Subsequent fermentation of the hydrolysate with the yeast *S. cerevisiae* indicated that maximum ethanol production was obtained with the microbe-fermenter-enhanced plant biomass hydrolysate (42.2 g/L), while the control phytoreactor achieved only 25.5 g/L. These results clearly indicate that the coupling of phytoremediation and bioethanol production can be a solution by which to eliminate the pollution problem and produce bioenergy. Theoretical bioethanol yields were calculated for six cultivars of the annual grass *S. bicolor*, which can adsorb HMs, especially Cd, from polluted soils [40]. The selected sorghum exhibited high Cd uptake (102 g/ha) and high bioethanol yield (6670 L/ha) and may be used for promising application.

Sweet sorghum cultivated in the presence of HMs was also the biomass explored by Vintilla et al. [136]. The fermentation process with *S. cerevisiae*, after the phytomass hydrolysis with cellulase, showed an increase in metal concentration in the liquid phase of

the fermented broth. The process of distillation was found to allow the extraction of HMs that remained in the residues and, at the same time, allow the digestate to be returned as a soil fertilizer. The treatment of bagasse then enabled the production of ethanol, biogas, and organic fertilizer through anaerobic digestion.

The relevance of using sorghum as an energy crop, linked to its use as a phytoremediation plant, has been proposed by Liu et al. [137], who identified 3 sorghum cultivars among 166 grown in cadmium-contaminated environments, and which could be used efficiently by coupling environmental remediation with bioethanol production. The identified cultivars showed a positive association between Cd accumulation and biomass growth and plant height. Such an agricultural system would be able to reduce the presence of Cd in the soil and achieve an economic return through the production of bioenergy.

Xiao et al. have recently implemented studies on sweet sorghum from phytoremediation of cadmium-contaminated soil [118]. The chemical pretreatment carried out with sulfuric acid degraded hemicelluloses and facilitated cellulose attackability, reaching high glucose yields, after enzymatic hydrolysis with a commercial enzyme mixture (Cellic® CTec2, Novozyme). Moreover, 96% of the Cd present in sorghum was enriched in the hydrolysate.

As previously reported, the perennial bioenergy crop *Miscanthus*, grown on soil containing HMs, was treated with protic ionic liquid [120], with the extraction of 99% of the main contaminants. The methodologies have indicated new possibilities in the integration of phytoremediation and biorefining, leading to environmental remediation, clean biofuel production and metal recovery, without the need to incinerate biomass. The recovered clean cellulose pulp was subjected to a hydrolytic process using commercial enzymes (CTec2, Novozymes), obtaining a glucose yield of 81.5%.

An efficient saccharification process has been carried out on *H. annuus* grown in several concentrations of metals (As, Cd, Co, Cr, Cu, Fe, Mn, Ni, Pb, and Zn) [132]. The consortium of fungi (*Pholiota adiposa* and *Armillaria gemina*) allowed for the obtainment of a significant conversion of sugars (61.7%), and consequently a good production of bioethanol (11.4 g/L), by using a seed culture of *S. cerevisiae*. The importance of appropriate pretreatment by which to improve biomass digestibility by enzymes and to convert hydrolysate to bioethanol has been discussed by Asad et al. [115]. Enzymatic hydrolysis was conducted on metal-free and trace metal-contaminated tobacco, birch, and willow biomass, using a commercial cellulase mixture of *Trichoderma reesei*. The results show that metals have little or no effect on biomass hydrolysis, and that monomeric sugars can be used for bioethanol production. An interesting study on a phytoremediation system identifies *Eichhornia crassipes* as a plant capable of effectively removing HMs, in particular, chromium [50]. Ethanol production was studied in a bioreactor using *S. cerevisiae* to ferment *E. crassipes* hydrolysate. The production of hydrolysate using biomass at different concentrations of chromium showed their possible exploitation to produce bioethanol or other biofuels, since the incidence of metals was not significant. Furthermore, the author suggested a chemical desorption process for the recovery of metals and their possible reuse. The biomass resulting from phytoremediation can also be valorized for the recovery of high added value and fine chemical compounds, as investigated by Angelini et al. [122]. The recovery process of soils polluted by polychlorinated biphenyls and HMs in southern Italy was carried outby using poplar. After a performic pre-treatment, the contaminants were found dissolved in the aqueous solution and removed, while the enzymatic hydrolysis of the biomass with commercial cellulose (*T. reseei*), allowed the obtainment of more than 80% glucose. In addition, high yields of levunilic acid and hydroxymethyl furfural were generated from the poplar residue through a two-step conversion, consisting of enzymatic hydrolysis and a reaction using $AlCl_3 \cdot 6H_2O/ H_2SO_4$. Poplar is a species widely studied for phytoremediation purposes. Transgenic poplars overexpressing a cytosolic glutamine synthetase were able to assimilate high amounts of nitrate, used as fertilizer [141]. The transgenic plants showed an increase in biomass compared with the control and

accumulated higher levels of proteins, chlorophylls, and total sugars, usable to obtain biofuels or fibers.

The idea of coupling phytoremediation and biorefinery has been pursued by Sotenko et al., using *Sinapis alba* and *H. annuus,* known as agricultural plants, to decontaminate soil polluted by nickel [133]. Up to 80% of the metal was removed from the biomass by aqueous extraction. Degradation by the fungus *Phanerochaete chrysosporium* was more effective on *S. alba*, which was less affected by Ni contamination, but extraction reduced the yield of available sugars and phenols. Instead, from *H. annus*, subjected to pretreatment and degradation, a higher final quantity of both sugars and phenols was obtained in the extracts.

Future studies on bioethanol and high-value-added products from biomass combined with the plant growth process of contaminated soils, must take into account many variables [142]. The choice of suitable plants is fundamental, considering the soil to be restored and the presence of specific contaminants. Further, high biomass yield and tolerance to pollutants are important factors for achieving adequate production of high value-added products.

*5.2. Biogas*

Fermentation is a widely used method by which to produce high added-value materials of industrial interest from lignocellulosic biomass. Although there are still few studies relating to the use of phytoremediation biomass in anaerobic digestion to produce biogas, which contains methane, butane, and propane [143], this technology represents an important challenge for the production of a biofuel.

Evaluation of the effect of HMs and substrate particle size on biogas production has been studied by Verma et al. [130]. The slurry from *E. crassipes* and *T. bispinnosa*, used for the remediation of toxics metals and electroplating industry effluent, showed a biogas production higher than the control plants grown in unpolluted water. The maximum yield of biofuels was achieved with 20% of diluted effluent, at 5 mm particle size and a substrate/inoculum ratio of 1:1. While using a higher concentration of effluents, the methane content in biogas was lower, due to the high concentration of metals that caused inhibition of the methanogenesis process. In particular, the production of biogas was 2430 c.c./100 g dm for *E. crassipes* and 1940 c.c./100 g dm for *T. bispinnosa*, with a methane content of 63.82% and 57.04%, respectively. Biogas production has also been investigated on canola, oat and wheat growth in Cd-contaminated soils [128]. The phytoremediation resulted in an accumulation of cadmium in the aerial parts, especially for canola. Regarding the fermentation processes conducted as diauxic growth, the biogas yields showed a clear improvement for all three biomasses analyzed in the presence of Cd compared with the control (159.37%, 179.23% and 111.34% of the control for canola, oat, and wheat, respectively).

Agricultural land in Belgium, moderately contaminated with trace elements, was remediated by *Z. mays* [138]. Cultivars were selected to optimize biogas production potential. The results indicate that there was no difference in biogas production between the plants grown on contaminated soil (215.23 $Nm^3$ $Mg^{-1}$ FW) and non-contaminated soil (194.4 $Nm^3$ $Mg^{-1}$ FW).

Studies on biogas production from five plants used for decontamination of soil containing Cu have been carried out by Cao et al. [129]. Among the selected species, *P. americana* L., *Z. mays* L., *B. napus* L., *Elsholtzia splendens Nakai*, and *Oenothera biennis* L., the latter required the shortest period of anaerobic digestion. Moreover, the biomass from phytoremediaton, with high Cu levels (100 mg $kg^{-1}$), showed an increase in methane content. The capacity of *Silybum marianum* and *H. annuus*, grown in the presence of trace metals, to produce biogas has also been studied [52]. Both species highlighted aerobic biodegradability, marked for the seeds able to produce more biogas (*S. marianum* 312–344 mL $g^{-1}$ and *H. annuus* 356–473 mL $g^{-1}$) than the vegetative parts (*S. marianum* 194–223 mL $g^{-1}$ and *H. annuus* 134–154 mL $g^{-1}$). Furthermore, the metals had no negative effect on energy production through anaerobic digestion or combustion.

A similar study on the use of four different plant species, *S. marianum*, *Piptatherum miliaceum*, *Nicotiana glauca* and *H. annuus*, for phytostabilization of soils contaminated by trace elements, investigated the possibility of performing aerobic and anaerobic degradation [53]. The best performance for biogas production through anaerobic digestion (inoculum obtained from a wastewater treatment plant) was performed by and *P. miliaceum*, whereas *N. glauca* showed the lowest biogas production, due to its high Pb content.

*H. annuus*, contaminated with HMs, was anaerobically digested following changes in the microbial community [95]. The performance of a continuous stirred tank reactor remained constant, and Cu and Zn were the only metals remaining in solution, whereas Cd, Pb, and Ni were precipitated with sulfur and hydroxide. The balance of microbial metabolism remained appropriate for anaerobic digestion, with an enrichment of methanobacteria during the process.

The possibility of using heavy-metal-contaminated rice straw for biogas production was studied by Xin et al. [135]. The biomass, adequately alkali pretreated (NaOH 6% *w/w*), released 86.95–97.69% of Cd, Pb, Cu, and Zn. At the same time, the total biogas and methane yields improved by 22.18% and 41.59%, compared with the control. Illumina sequencing analysis showed an enrichment of fermentative bacteria and methanogenic archaea, which are necessary to produce biogas and release HMs. The *P. vittata* biomass, recovered after a phytoremediation experiment of As-contaminated soil, was used to produce biomethane [114]. The polluted biomass was subjected to ethanol extraction, followed by anaerobic digestion, using mesophilic microorganisms obtained from the anaerobic digestion of food waste. The coupled technique made it possible to remove 98% of the As, recovered in the digestate supernatant, while producing methane with yields comparable with the use of uncontaminated biomass.

*P. vittata* was also the species examined as a heavy metal hyperaccumulator, along with *Solanum nigrum*, *S. alfredii*, *P. Americana* and the Cu-accumulator *Elsholtzia haichowensis*, by Wang et al. [131]. Their results show that Cu and Mn (at a concentration of 5000–10,000 mg/kg) improved biogas and methane production, especially for *E. haichowensis*, while the presence of Zn (>500 mg/kg) drastically reduced the biogas yield for *S. alfredii*. Cd, Pb, and As presence had no influence on the anerobic degradation and subsequent biogas production of *S. nigrum* (135. 9 mL/g), *S. alfredii* (238.7 mL/g), *P. vittata* (106.8 mL/g), *P. americana* (129.5 mL/g) and *E. haichowensis* (259.2 mL/g).

The presence of mercury in the anaerobic degradation of *Lepidium sativum* L. and *Mentha spicata* revealed that a high concentration of HM was found to be detrimental for biogas production, due to its bacteriostatic action [134]. When phyto plants, grown in the presence of Hg, were digested in continuously stirred batches in the presence of canola oil–sulfide polymer, used as an intermediate treatment, the metal was extracted from the digestate with an efficiency of 40–50%. The combination of anaerobic digestion with the polymer extraction offers an important opportunity to use the species for the phytoremediation of Hg and its simultaneous recovery. Recently, Paulo et al. 139] examined the growth and biogas production of *Z. mays* [34]. The plant was grown in Zn and Cd-contaminated soil and coupled with the mycorrhizal fungus *Rhizophagus irregularis* and the rhizobacteria *Cupriavidus* sp. strain 1C2. The combination strategy permitted a removal of 0.77% and 0.13% of the Cd and Zn present in the soil with an increase in plant yield (9%). Interestingly, the study on biogas and methane production clearly indicates that metals do not prevent anaerobic biodegradation of biomass. In fact, the biomethane production levels detected were 183 and 178 mL of $CH_4$ $g^{-1}$ for the plant grown in uncontaminated and contaminated soil, respectively.

The growing interest demonstrated by the studies aimed at the reuse of exploited plants in land reclamation confirms the great potential for biogas production. Further feasibility studies should consider the technical aspects of biofuel production, proper treatment and removal of contaminants and microbial community dynamics.

*5.3. Biodiesel*

Historically, the production of biodiesel from plant feedstocks dates back to the mid-nineteenth century. Biofuel is, in fact, renewable, low cost, and biodegradable with a high calorific value and a low sulfur and aromatic content. The most used process by which to make vegetable oil less viscous is the process of transesterification, usually catalyzed by alcohols or acids/bases [144].

Some studies focus on the possible coupling of plant species recognized as valuable for phytoremediation and their ability to produce biodiesel, effectively increasing the added value of the biofuel.

Rheay et al. [145] analyzed the ability of *C. sativa* L. to phytoremediate soils contaminated by toxic metals, radionuclides, and organic pollutants and its versatility in use for biofuels and biodiesel production. The oil content in hemp seeds is between 25 and 35% ($w/w$) and it has similar physicochemical properties to those used for biodiesel mixture.

A species extensively studied for its interesting role in the phytoremediation of heavy metal-contaminated soils is *Ricinus communis* L. The castor bean plant has been used as a multipurpose crop for phytoremediation, phytostabilization and revegetation of soils contaminated by waste disposed in contaminated peri-urban areas [146]. The plant is important for its high concentration of ricinoleic acid (12-hydroxy-9-octadecenoic acid), which constitutes 89% of the oil, and other fatty acids, including linoleic acid, oleic acid, stearic acid, palmitic acid, linolenic acid and eicosanoic acid [147]. Due to the high oil content in the seeds, it can be conveniently used for biodiesel production [148]. Comparable characteristics have also been found for the cannula species *Jatropha curcas* [149]. *J. curcas* possesses rapid growth and phytoremediation characteristic in the presence of HMs, combined with a high oil content. Interestingly, blending castor oil with jatropha oil has enabled the production of biodiesel with better fuel characteristics [150]. Among the species investigated as potential feedstock to produce biodiesel, *Echinochloa* crus-galli has been explored for its facility to grow in heavy-metal-contaminated areas and its capacity to translocate the pollutants from contaminated sites. At the same time, the feedstock can be used for the direct conversion of its seeds into biodiesel [151].

To date, there are no studies concerning the production of biodiesel from plants through phytoremediation carried out via transesterification mediated by biocatalysts, even if enzymatic catalysis is used in the production of biodiesel.

Compared with the transesterification of triglycerides with chemical catalysts, the production of biodiesel by enzymatic catalysis using lipases presents the ease of separation of the products, the reduction of pollutant and wastewater treatment requirements, and the total absence of side reactions [152]. Similar negative aspects to the use of enzymatic transesterification are the longer reaction time requirement than chemical-catalyzed techniques and the risk of enzyme inactivation due to its sensitivity to the solvents [153].

Therefore, feasibility studies are necessary to verify the compatibility of biodiesel production from plants and from weeds not associated with the emission of harmful and prohibited metals and contaminants [154].

The use of biocatalysts in biodiesel production processes appears to be a more sustainable alternative to reduce both environmental and health risks, although high production costs and some problems related to technical processes make these not yet adequately explored. Biodiesel obtained through enzymatic transesterification is capable of producing high quality biofuel, lowering the toxicity issues associated with chemical catalysis and its removal from the reaction mixture [155].

## 6. Future Perspective

Phytoremediation of polluted sites is an eco-friendly technique for addressing soil contamination and provides a beneficial solution for soil remediation and a potential sustainable practice. Moreover, this technology, combined with the production of biomass useful for biofuel production using green techniques, is strongly required by the goals of the European Green Deal [98]. Integrated phytoremediation assessment requires quantification

of the real performance at a site-specific condition, and an accounting for biochemical interactions, depending on the landscape, soil properties, vegetation species, weather conditions, and pollutant types.

In addition, the biochemical methodologies applied require an active scientific, economic and legislative integration for a complete analysis. Research and policy development to promote this form of management is lacking. There is a need for the policy maker to enact clear laws on phytoremediation that provide for biomass management after phytoremediation and promote recycling (using environmentally friendly methods) of biomass environmental hazard [156].

The development of a dedicated and harmonized European legal regime that addresses sustainable soil management and sets the tone for a coordinated approach to soil remediation, including non-conventional techniques, is essential [98].

The evaluation of the economic aspects of phyto techniques for the treatment of polluted lands should include consideration of various options for generating revenue through the collection and marketing of contaminated biomass. These include the investment costs necessary to implement phytoremediation and the revenues deriving from the exploitation of biomass. The long-term project evaluation can then be carried out through different phases, identifying the economic hotspots [156].

However, there are few studies available that concern these financial aspects, though the investments it certainly requires are nevertheless potentially recoverable. In fact, it is expected that biomass from phytoremediation can be used to produce energy and high added-value products, such as essential oils, the demand for which will exceed five trillion US dollars [3]. In addition to this, due account must be taken of the reduction of virgin raw materials to produce energy, and the reduction of greenhouse emissions [60] The economic and feasibility studies must also include sustainability criteria, such as the soil remediation service, the overall preservation of ecosystem services and the protection of land with high biodiversity.

## 7. Conclusions

Currently, pervasive environmental pollution and increasing renewable energy demand are the most important challenges at the global level. Most research efforts towards the implementation of effective and environmentally friendly remediation techniques are focused on phytoremediation, which is considered one of the most promising technologies for the decontamination of polluted lands. This exciting approach to phyto technologies, centered on the exploitation of polluted marginal soils for production of energy crops with high decontamination potential, represents a novel frontier that strongly encourages interdisciplinary studies across economical, agronomic, ecological, chemical, environmental engineering and biocatalysis sciences. Particularly important is the choice of suitable species that, while contributing to the restoration of polluted areas, can allow metals recovery, an attractive and valuable resource from post-phytoremediation biomass.

The reutilization of bio-waste from phytoremediation with the least environmental impact, is one of the main objectives of numerous studies, due to the enormous potential that derives from the conversion of the lignocellulosic fraction into biofuels and high value-added products. To achieve these goals, studies that can couple phytoremediation processes and biocatalytic methods for biomass conversion with the least environmental impact have been performed. The resulting bioenergy production could significantly contribute to the global demand for renewable energy and to a complete and fully circular economy. Further in-depth studies are undoubtedly necessary to structure the whole process and to promote it at field and industrial level. The choice of the most suitable species is crucial, and genetic engineering can also increase the phytoremediation capabilities of the selected plants that are often exposed to a myriad of pollutants (HMs, metalloids, hydrocarbons, insecticides and new, emerging contaminants). Adequate agronomic practices need also to be implemented to mitigate the impact of multiple toxicants and maximize plant biomass yields and decontamination efficiencies. More efforts by plant biologists and geneticists

must also be addressed to target the transit and compartmentalization of toxic compounds in the vegetal organism in a more predictable way, in order to allow well designed industrial biorefinery processes for the maximization of resource recovery with zero waste outputs and zero risk of secondary pollution.

Future prospects and advancements of "sustainable phytoremediation" that will also open to a wider popular acceptance and far-reaching profits will be strictly linked to the development of hyperaccumulator species provided with novel features that make them more effective for remediation and bioenergy production issues. It is worth noting that hyperaccumulators are usually slow growing, small plants whose distribution is very locally limited, with several regions characterized by a really exiguous number of these kind of species. Actual intensive research studies have aimed at the development of fast-growing energy crops with deep roots, high biomass production and high capacity of contaminant accumulation are supported by important international funding. Moreover, the translation to the field of the developed remediation practices in various areas with different physic and geochemical characteristics, and which are affected by different types of pollution, will allow the development of important remediation strategies applicable on a wide spectrum of contaminated sites. Fundamental tasks of the recently and sizably funded projects are those that pursue the development of suitable technological solutions by which to optimize the exploitation of post-phytoremediation biomass to produce biofuels that are clean from contaminants, with the aim of eliminating the risks of secondary pollution. In this regard, of particular importance will be the implementation of novel and more effective methodologies aimed at the recovery of metal from plant biomass. This will also ensure high profitability margins, especially in the case of rare and/or precious ores with high added value for high tech industrial applications.

Finally, to move the "sustainable phytoremediation" strategy from dream to reality, adequate analyses at socio-economic levels are necessary in order to evaluate potential impacts on the market of the different phyto-derived products and to determine any occupational effects.

**Author Contributions:** Conceptualization, E.I. and L.M. (Loredana Marcolongo); writing—review and editing, E.I, E.C., L.M. (Loredana Marcolongo) and L.M. (Luigi Mandrich). All authors have read and agreed to the published version of the manuscript.

**Funding:** This research was funded by the Italian "Ministero dell'Università e della Ricerca Scientifica"; Industrial Research Project "Technologies and processes for pollutants reduction and contaminates sites bioremediation with commodities recovery and totally green energy production-TARANTO" PONARS01_00637.

**Institutional Review Board Statement:** Not applicable.

**Informed Consent Statement:** Not applicable.

**Data Availability Statement:** The data presented in this study are available in PubMed, and all the referment used are reported in the References section.

**Acknowledgments:** We are grateful to Bernard Loeffler and Francesca Varrone for their excellent editing assistance.

**Conflicts of Interest:** The authors declare no conflict of interest.

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
