# Peer review of "Moving towards Biofuels and High-Value Products through Phytoremediation and Biocatalytic Processes"

_catalysts, doi:10.3390/catal14020118_

Round 1
Reviewer 1 Report
Comments and Suggestions for Authors
Authors are advised to revise the manuscript accordingly.

Author Response
We report point by point our answers (in italic) to the referee comments (in bold).
Major Comments
Abstract should be rewritten highlighting the novelty of the present review and with emphasis on what the review is offering to the readers.
We have rewritten the abstract; we hope that now it will be more appealing.
Provide proper “keywords”. For example: “Biorefinery” and “Circular economy” has been used only 5 times.
We have substituted the keywords “Biorefinery” and “Circular economy” with “Biofuel” and “Integrated techniques”.
Briefly present the financial related statistics apart from US.
We have reported financial data about the founding estimated by the European Community by 2048. Page 2, line 82-84.
Last paragraph of the introduction should highlight the core highlights presented in the review and the novelty.
As request from the reviewer, we have added a new part indicating in more details the highlights of the review. Page2-3, lines 122-142.
Can go through few recent articles. https://doi.org/10.1016/j.bcab.2023.102845; https://doi.org/10.3390/fermentation9110990; https://doi.org/10.3390/plants12081653.
We have replaced and used new and recent references, as suggested by the reviewer: references number 3 and 106.
Tabulate various pretreatment technologies involved as stated in section 4 along with their applicability, advantages, and disadvantages. List out the possible limitations, as use of heavy metals based plant biomass shows major drawbacks in real time applications as per literature.
As suggested by the reviewer, we added a new part at page 12, lines 549-565. Moreover, we inserted a new Table, listed as table 2, in which we reported the applicability, advantages, and disadvantages of the pretreatment technologies reported in section 4. Page 15, line 687.
Include more recent literature, only 54 out of 155 were from past three years (on or after 2020).
As suggested by the reviewer, we replaced 4 old references with others more recent (reference number 1, 3, 80 and 106. Moreover, we inserted a new reference (number 156).
Minor Comments
Line 204: Already abbreviated in line 34. Ok, we corrected it.
Line 210: Already abbreviated in line 35. Ok, we corrected it.
Line 113, 252, 452: Already abbreviated in line 36. Ok, we corrected it.
Line 39: No need to abbreviate European Environmental Agency. State in full. Yes, we do it
Line 48: “25-30 billion” in what? Ok, we added “billion dollars.”
Table 2: “H2SO4”. Ok, we corrected it.
Line 613, 614: “42.2 g/L” and “25.5 g/L”. Ok, we corrected it.
Scientific names should be stated in full upon first mention, after which genus should be abbreviated and species should be stated in full. For example Zea mays (Z. mays). Check the same for other scientific names throughout the manuscript.
Ok, we have controlled and corrected in the text all the scientific names and reported the correct abbreviation for them
Remark
The review presents proper background information but lacks in depth analysis or interpretation of recent literature. Figures presented are general and does not offer anything significant.
The topic that we have chosen for this review is very broad, and it has been recently enriched by a growing literature. We wanted to give a general overview that would stimulate readers in-depth analysis into more specific argument of the topic. The figures that we have prepared, are with the aim to visually summarize what is reported in the text. We rewritten the paragraph “future perspective” and the “Conclusion” to give a better analysis of the reported data.
Reviewer 2 Report
Comments and Suggestions for Authors
The review article titled "Moving towards biofuels and high value products through phytoremediation and biocatalytic processes" focuses on the utilization of phytoremediation and biocatalytic processes in environmental remediation and the production of biofuels and high-value products. The article elaborates on the mechanisms of phytoremediation, the selection of plants for this purpose, and the valorization of by-products from these processes. The article is well structured, however there are some queries and suggestions as below:
1. Are the figures (1, 2 and 3) are original? If no, please cite the relevant work and provide the copyright details.
2. Section 4, the authors discussed the pretreatment methods of lignocellulosic materials, the authors have mentioned many references in this section, I suggest to providing a table to represent the work done on the pretreatment context.
3. More emphasis on the long-term environmental impacts of these methods, such as the effect on biodiversity and ecosystem balance, would be valuable. This could include discussions on any potential risks associated with the accumulation of biomass or by-products.
4. I suggest to write in more details about the future prospects and potential advancements in the field of phytoremediation and biocatalysis?
5. Further exploration of the technological challenges and economic feasibility of scaling up these methods for widespread industrial use would be useful. This includes analysis of the required investments, potential returns, and economic incentives.
6. What are the authors' insights on the policy and regulatory changes needed to facilitate the adoption of these methods on a larger scale? Please provide insights about this.
7. Check the spelling in page 13, line 557, “Figure 3.S chematic illustration”
Comments on the Quality of English LanguageEnglish language revision is required.
Author Response
We report point by point our answers (in italic) to the referee comments (in bold).
Comments and Suggestions for Authors
The review article titled "Moving towards biofuels and high value products through phytoremediation and biocatalytic processes" focuses on the utilization of phytoremediation and biocatalytic processes in environmental remediation and the production of biofuels and high-value products. The article elaborates on the mechanisms of phytoremediation, the selection of plants for this purpose, and the valorization of by-products from these processes. The article is well structured, however there are some queries and suggestions as below:
- Are the figures (1, 2 and 3) are original? If no, please cite the relevant work and provide the copyright details.
The figures 1, 2 and 3 were prepared by us, being schematic representation they could be similar to others, but we have prepared the figure following the text of our review.
- Section 4, the authors discussed the pretreatment methods of lignocellulosic materials, the authors have mentioned many references in this section, I suggest to providing a table to represent the work done on the pretreatment context.
As suggested by the reviewer, we inserted a new Table, listed as table 2, in which we reported the applicability, advantages, and disadvantages of the pretreatment technologies reported in section 4. Page 15, line 687.
- More emphasis on the long-term environmental impacts of these methods, such as the effect on biodiversity and ecosystem balance, would be valuable. This could include discussions on any potential risks associated with the accumulation of biomass or by-products.
We report a referment about it in the “Discussion” section. Page 23
- I suggest to write in more details about the future prospects and potential advancements in the field of phytoremediation and biocatalysis?
We report a referment about it in the “Discussion” section. Page 23
- 5. Further exploration of the technological challenges and economic feasibility of scaling up these methods for widespread industrial use would be useful. This includes analysis of the required investments, potential returns, and economic incentives.
We rewritten the paragraph “future perspective” and the “Conclusion” in which we reported information about the reviewer request.
- What are the authors' insights on the policy and regulatory changes needed to facilitate the adoption of these methods on a larger scale? Please provide insights about this.
We rewritten the paragraph “future perspective” and the “Conclusion” in which we reported information about the reviewer request.
- Check the spelling in page 13, line 557, “Figure 3.S chematic illustration”
Ok, we corrected it.
Comments on the Quality of English Language: English language revision is required.
We revised some parts of the manuscript; they are as new track change. The English editing was further revised by Bernard Loeffler and Francesca Varrone, our editing assistants. As their request, in the revised version of the review we inserted in the “Acknowledges” section both editing assistant.